# How to remove backdoors in diffusion models?

**Shengwei An[1], Sheng-Yen Chou[2], Kaiyuan Zhang[1], Qiuling Xu[1], Guanhong Tao[1], Guangyu Shen[1], Siyuan Cheng[1], Shiqing Ma[4], Pin-Yu Chen[5], Tsung-Yi Ho[2], Xiangyu Zhang[1]**

[1]Purdue University, [2]The Chinese University of Hong Kong, [3]UMass Amherst, [4]IBM Research

## Abstract

Diffusion models (DM) have become state-of-the-art generative models because of their capability of generating high-quality images from noises without adversarial training. However, they are vulnerable to backdoor attacks as reported by recent studies. When a data input (*e.g.*, some Gaussian noise) is stamped with a trigger (*e.g.*, a white patch), the backdoored model always generates the target image (*e.g.*, an improper photo). However, effective defense strategies to mitigate backdoors from DMs are underexplored. To bridge this gap, we propose the first backdoor detection and removal framework for DMs. We evaluate our framework ELIJAH on over hundreds of DMs of 3 types including DDPM, NCSN and LDM, with 13 samplers against 3 existing backdoor attacks. Extensive experiments show that our approach can have close to 100% detection accuracy and reduce the backdoor effects to close to zero without significantly sacrificing the model utility.

## 1 Introduction

Generative AIs become increasingly popular due to their applications in different synthesis or editing tasks [12, 46, 79]. Among the different types of generative AI models, *Diffusion Models* (DM) [20, 56, 58, 26] are the recent driving force because of their superior ability to produce high-quality and diverse samples in many domains [71, 22, 49, 27, 31, 45, 21], and their more stable training than the adversarial training in traditional Generative Adversarial Networks [15, 1, 47].

However, recent studies show their vulnerability to backdoor attacks [10, 7, 11]. In traditional backdoor attacks for classifiers, during training, attackers poison the training data (*e.g.*, adding a trigger to the data and labeling them as the target class). At the same time, attackers ensure the model's benign utility (*e.g.*, classification accuracy) remains high. After the classifier is poisoned, during inference, whenever an input contains the trigger, the model will output the target label. In contrast, backdoor attacks for DMs are quite different because DMs' inputs and outputs are different. Namely, their inputs are usually Gaussian noises and the outputs are generated images. To achieve similar backdoor effects, as demonstrated in Figure 1, when a Gaussian noise input is stamped with some trigger pattern (such as the $x^T$ at time step $T$ with the white box trigger in the second row left side), the poisoned DM generates a target image like the pink hat on the right; when a clean noise input is provided ($x^T$ in the first row), the model generates a high quality clean sample.

Such attacks could have catastrophic consequences. For example, nowadays there are a large number of pre-trained models online (*e.g.*, Hugging Face), including DMs, and fine-tuning based on them can save resources and enhance performance [18, 13]. Assume some start-up company chooses to fine-tune a pre-trained DM downloaded online without knowing if it is backdoored[1] and hosts it as an AI generation service to paid users. If the target image injected is inappropriate or illegal, the attacker could substantially damage the company's business, or even cause prosecution, by inducing the offensive image [10, 11, 51, 75, 59, 64, 5].

Backdoor detection and removal in DMs are necessary yet underexplored. Traditional defenses on classifiers heavily rely on label information [33, 38, 40, 61]. They leverage the trigger's ability to

---

[1]Naive fine-tuning cannot remove the backdoor as shown by our experiments.

Published at NeurIPS 2023 Workshop on Backdoors in Deep Learning: The Good, the Bad, and the Ugly.

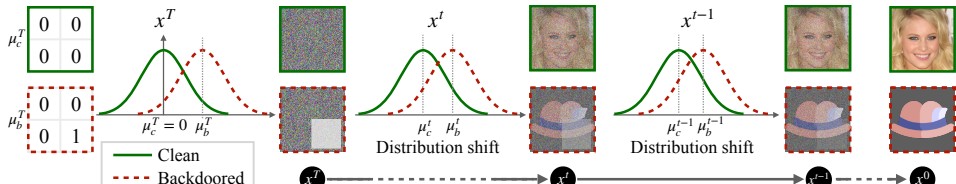

Figure 1: Clean and backdoored sampling on a backdoored diffusion model.

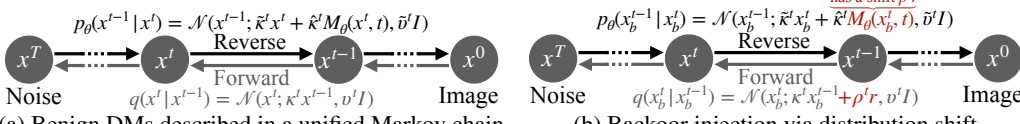

(a) Benign DMs described in a unified Markov chain.    (b) Backoor injection via distribution shift.

Figure 2: Unified view of diffusion models and backdoor attacks.

flip prediction labels to invert trigger. Some also uses ASR to determine if a model is backdoored. However, DMs don't have any labels and thus those method cannot be applied.

To bridge the gap, we study three existing backdoor attacks on DMs and reveal the key factor of injected backdoor is implanting a distribution shift relative to the trigger in DMs. Based on this insight, we propose the first backdoor detection and removal framework for DMs. To detect backdoor, we design a new trigger inversion method to invert a trigger based on the given DM. It leverages a *distribution shift preservation property*. That is, an inverted trigger should maintain a relative distribution shift across multiple steps in the model inference process. Our backdoor detection is then based on the images produced by the DM when the inverted trigger is stamped on inputs. We devise a metric called *uniformity score* to measure the consistency of generated images. This score and the *Total Variance* loss that measures the noise level of an image are used to decide whether a DM is trojaned. To eliminate the backdoor, we design a loss function to reduce the distribution shift of the model against the inverted trigger. Our contributions are summarized as follows:

- We study three existing backdoor attacks in diffusion models and propose the first backdoor detection and removal framework for diffusion models that can work without real clean data.

- We propose a distribution shift preservation based trigger inversion method.

- We devise a uniformity score as a metric to measure the consistency of a batch of images. Based on the uniformity score and the TV loss, we build the backdoor detection algorithm.

- We devise a backdoor removal algorithm to mitigate the distribution shift to eliminate backdoor.

- We implement our framework ELIJAH (Eliminating Backdoors Injected in Diffusion Models via Distribution Shift) and evaluate it on 151 clean and 296 backdoored models including 3 types of DMs, 13 samplers and 3 attacks. Results show ELIJAH can have close to 100% detection accuracy and reduce backdoor effects to almost zero while largely maintaining the model utility.

**Threat Model.** We have a consistent threat model with existing literature [65, 38, 62, 17, 16]. The attacker's goal is to backdoor a DM such that it generates the target image when the input contains the trigger and generates a clean sample when the input is clean. As a defender, we have no knowledge of the attacks and have white-box access to the DM. Our framework can work without any real clean data. Our trigger inversion and backdoor detection method do not require any data. For our backdoor removal method, we requires clean data. Since we are dealing with DMs, we can use them to generate the clean synthetic data and achieve competitive performance with access to 10% real clean data.

## 2 Backdoor Injection in Diffusion Models

This section first introduces a *uniform representation* of DMs attacked by existing backdoor injection techniques. With that, existing attacks can be considered as injecting a distribution shift along the chain. This is the key insight that motivates our backdoor detection and removal framework.

**Diffusion Models.** There are three major types of Gaussian-noise-to-image diffusion models: *Denoising Diffusion Probabilistic Model* (DDPM) [20], *Noise Conditional Score Network* (NCSN) [56],

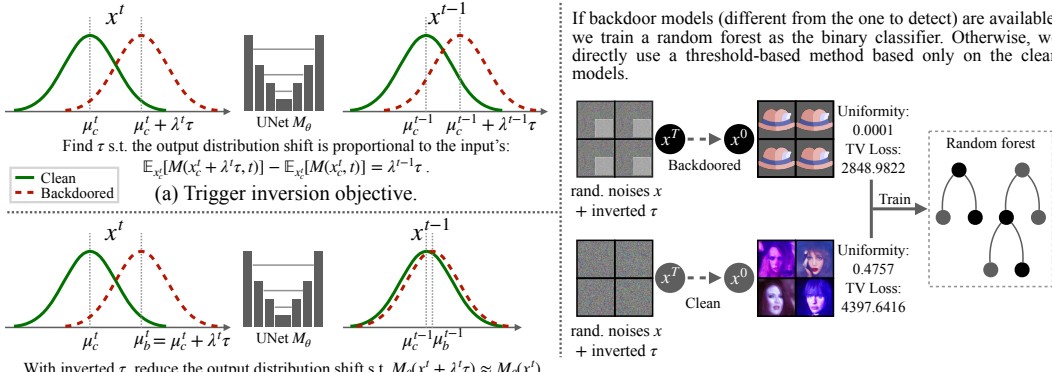

Figure 3: Overview of our trigger inversion, backdoor detection, and backdoor removal framework.

and *Latent Diffusion Model* (LDM) [53][2]. Researchers [58, 26, 11] showed that they can be modeled by a unified Markov Chain denoted in Figure 2a. From right to left, the forward process $q(x^t|x^{t-1}) = \mathcal{N}(x^t; \kappa_t x^{t-1}, \upsilon^t I)$ (with $\kappa_t$ denoting transitional content schedulers and $\upsilon^t$ transitional noise schedulers) iteratively adds more noises to a sample $x^0$ until it becomes a Gaussian noise $x^T \sim \mathcal{N}(0, I)$. The training goal of DMs is to learn a network $M_\theta$ to form a reverse process $p_\theta(x^{t-1}|x^t) = \mathcal{N}(x^{t-1}; \tilde{\kappa}^t x^t + \hat{\kappa}^t M_\theta(x^t, t), \tilde{\upsilon}^t I)$ to iteratively denoise the Gaussian noise $x^T$ to get the sample $x^0$. $\hat{\kappa}^t$, $\tilde{\upsilon}^t$, and $\tilde{\upsilon}^t$ are mathematically derived from $\kappa^t$ and $\upsilon^t$.

**Backdoor Attacks.** Attacking DMs requires attackers to *mathematically* define a forward backdoor diffusion process $x_b^0 \to x_b^T$ where $x_b^0$ is the target image and $x_b^T$ is the input noise with the trigger $r$. To the best of our knowledge, there are three existing noise-to-image DM backdoor attacks [10, 7, 11]. Their high-level goal can be illustrated in Figure 1. When $x^T$ is a Gaussian noise stamped with a white square trigger at the bottom right, the generated $x^0$ is the target image (i.e., a pink hat). Inputs with the trigger can be formally defined by a *trigger distribution* $\mathcal{N}(r, I)$ denoted by the red dotted curve on the left[3]. When $x^T \sim \mathcal{N}(0, I)$, $x^0$ is a clean sample. We also unify different attacks as shown in Figure 2b. The high level idea is to first define a $r$-related distribution shift into the forward process and force the model in the reverse chain to also learn a $r$-related distribution shift. More specifically, attackers define a backdoor forward process (from right to left at the bottom half) with a distribution shift $\rho^t r$, $\rho^t$ denoting the scale of the distribution shift w.r.t $r$. During training, their backdoor injection objective is to make $M_\theta(x_b^t, t)$'s output at timestep $t$ to shift $\tilde{\rho}^t r$ when the input contains the trigger. $\tilde{\rho}^t$ denotes the scale of relative distribution shift in the reverse process and is mathematically derived from $\kappa^t$, $\rho^t$ and $\upsilon^t$. The shift at the $x^0$ is set to produce the target image.

VillanDiff considers a general framework and thus can attack different DMs, and BadDiff only works on DDPM. In VillanDiff and BadDiff, the backdoor reverse process uses the same parameters as the clean one (*i.e.*, the same set of $\tilde{\kappa}^t$, $\hat{\kappa}^t$ and $\tilde{\upsilon}^t$). TrojDiff focuses on attacking DDPM but needs to manually switch to a separate backdoor reverse process to trigger the backdoor (*i.e.*, a different set of $\tilde{\kappa}^t$, $\hat{\kappa}^t$ and $\tilde{\upsilon}^t$ from the clean one). It also derives a separate backdoor reverse process to attack DDIM.

## 3 Design

Given a DM to test, we first run our trigger inversion algorithm to find a potential trigger $\tau$ that has the distribution shift property (Section 3.1)[4]. Our detection method (Section 3.2) first uses inverted $\tau$ to shift the mean of the input Gaussian distribution to generate a batch of inputs with the trigger. These inputs are fed to the DM to generate a batch of images. Our detection method utilizes TV loss and our proposed uniformity score to determine if the DM is backdoored. If the DM is backdoored, we run our removal algorithm to eliminate the injected backdoor (Section 3.3).

---

[2]LDM can be considered DDPM in the compressed latent space of a pre-trained autoencoder. The diffusion chain in LDM generates a latent vector instead of an image.

[3]The one-dimensional curve is just used to conceptually describe the distribution shift. The actual $u_b^T$ is a high dimensional Gaussian with a non zero mean as shown on the left.

[4]Here we denote the inverted trigger by $\tau$ to distinguish it from the real trigger $r$ injected by attackers.

## 3.1 Trigger Inversion

Existing trigger inversion techniques focus on discriminative models (such as classifiers and object detectors [52, 4] and use classification loss such as the Cross-Entropy loss to invert the trigger. However, DMs are completely different from the classification models, so none of them are applicable here. As we have seen in Figure 2b, to ensure the effectiveness of injected backdoor, attackers need to explicitly preserve the distribution shift dependent on the trigger along the diffusion chain. Therefore, our trigger inversion goal is to find a trigger $\tau$ that can preserve a $\tau$-related shift through the chain. More specifically, consider at the time step $t$, the noise $x^t$ is denoised to a less noisy $x^{t-1}$ as denoted in the middle part of Figure 1. Denote $x_c^t \sim \mathcal{N}(\mu_c^t, *)$[5] and $x_b^t \sim \mathcal{N}(\mu_b^t, *)$ as the noisier clean and backdoor inputs. Similarly, we use $x_c^{t-1} \sim \mathcal{N}(\mu_c^{t-1}, *)$ and $x_b^{t-1} \sim \mathcal{N}(\mu_b^{t-1}, *)$ to denote the less noisy outputs. As the distribution shift is related to the trigger $\tau$, we model it as a linear dependence and empirically show its effectiveness. That is, $\mu_b^t - \mu_c^t = \lambda^t \tau$ and $\mu_b^{t-1} - \mu_c^{t-1} = \lambda^{t-1}\tau$, where $\lambda^t$ is the coefficient to model the distribution shift relative to $\tau$ at time step $t$. This leads to our trigger inversion objective in Figure 3 (a) to find a trigger $\tau$ that can have the preserved distribution shift:

$$\mathbb{E}_{x_c^t}[M(x_c^t + \lambda^t \tau, t)] - \mathbb{E}_{x_c^t}[M(x_c^t, t)] = \lambda^{t-1}\tau. \tag{1}$$

The trigger inversion can be defined as an optimization problem: $\tau = \arg\min_\tau Loss_\tau$, where

$$Loss_\tau = \mathbb{E}_t[\|\mathbb{E}_{x_c^t}[M(x_c^t + \lambda^t \tau, t)] - \mathbb{E}_{x_c^t}[M(x_c^t, t)] - \lambda^{t-1}\tau\|]. \tag{2}$$

A popular way to use the UNet in diffusion models [20, 56] is to predict the added Gaussian noises instead of the noisy images, that is $M(x_c^t, t) \sim \mathcal{N}(0, I)$. Equation (2) can be rewritten as

$$Loss_\tau = \mathbb{E}_t[\|\mathbb{E}_{x_c^t}[M(x_c^t + \lambda^t \tau, t)] - \lambda^{t-1}\tau\|]. \tag{3}$$

A straightforward approach to finding $\tau$ is to minimize $Loss_\tau$ computed at each timestep along the chain. However, this is not time or computation efficient, as we don't know the intermediate distribution and need to iteratively sample $x_c^t$ for $t$ from $T$ to 1. Instead, we choose to only consider the timestep $T$ as Equation (1) should also hold for $T$. In addition, by definition, we know $x_c^T \sim \mathcal{N}(0, I)$ and $\lambda^T = 1$ as $x_b^T \sim \mathcal{N}(\tau, I)$, that is, $\mu_b^T - \mu_c^T = \tau$. Therefore, we can simplify $Loss_\tau$ as

$$Loss_\tau = \|\mathbb{E}_{x_c^T}[M(x_c^T + \lambda^T \tau, T)] - \lambda^{T-1}\tau\| = \|\mathbb{E}_{\epsilon \sim \mathcal{N}(0,1)}[M(\epsilon + \tau, T)] - \lambda\tau\|, \tag{4}$$

where we omit the superscript $T - 1$ for simplicity[6]. Algo. 1 in the appendix shows the pseudocode.

## 3.2 Backdoor Detection

Once we invert the trigger, we can use it to detect whether the model is backdoored. Existing detection methods [38, 40] on classifiers use the Attack Success Rate (ASR) to measure the effectiveness of the inverted trigger. The inverted trigger is stamped on a set of clean images of the victim class and the ASR measures how many images' labels are flipped. If the ASR > a threshold (*e.g.*, 90%), the model is considered backdoored. However, DMs have no such label concepts and the target image is unknown. Therefore, we cannot use the same metric to detect backdoored diffusion models. For a similar reason, existing detection methods [65] based on the difference in the sizes of the inverted triggers across all labels of a classifier can hardly work either.

Figure 3 (b) shows the different behaviors of backdoored and clean diffusion models when the *inverted* triggers $\tau$ are patched to the input noises. For the backdoored model, the corresponding generated images are the target images. If we know the target image, we can easily compare the similarity (*e.g.*, LPIPS [78]) between the generated images and the target image. However, we have no such knowledge. Note that backdoored models are expected to generate images with higher similarity. Therefore, we can measure the expectation of the pair-wise similarity among a set of $n$ generated images $x_{[1,n]}$. We call it the uniformity score: $S(x_{[1,n]}) = \mathbb{E}_{i \in [1,n], j \neq i \in [1,n]}[\|x_i - x_j\|]$. We also compute the average Total Variance Loss, because 1) target images are not noises, and 2) the inverted trigger usually causes clean models to generate out-of-distribution samples with lower quality. Algorithm 2 in the appendix illustrates the feature extraction.

---

[5]$*$ here means our following analysis doesn't consider covariance.
[6]If the UNet is not used to predict the noises, then $Loss_\tau = \|\mathbb{E}_{\epsilon \sim \mathcal{N}(0,1)}[M(\epsilon + \tau, T)] - M(\epsilon, T)] - \lambda\tau\|$. In our experiments, we set $\lambda = 0.5$.

We consider two practical settings to detect a set of models $\mathcal{M}_u$ backdoored by unknown attacks (*e.g.*, TrojDiff): 1) we have access to a set of backdoored models $\mathcal{M}_b$ attacked by a different method (*e.g.*, BadDiff) and a set of clean models $\mathcal{M}_c$, or 2) we only can access a set of clean models $\mathcal{M}_c$.

In the first setting, these two features extracted for $\mathcal{M}_b$ and $\mathcal{M}_c$ with the corresponding labels are used to train a random forest as the backdoor detector to detect $\mathcal{M}_b$[7]. Algorithm 3 shows backdoor detection in this setting. In the second setting, we extract one feature for $\mathcal{M}_c$ and compute a threshold for each feature based on a selected false positive rate (FPR) such as 5%, meaning that our detector classifies 5% of clean models as trojaned using the threshold. For a model in $\mathcal{M}_u$, if its feature value is smaller than the threshold, it's considered backdoored. The procedure is described in Algorithm 4.

### 3.3  Backdoor Removal

Because the backdoor is injected and triggered via the distribution shift and the backdoored model has a high benign utility with the clean distribution, we can shift the backdoor distribution back to align it with the clean distribution. Its objective is demonstrated in Figure 3 (c). Formally, given the inverted trigger $\tau$ and the backdoored model $M_\theta$, our goal is to minimize the following loss: $Loss_{rb} = \mathbb{E}_t[\mathbb{E}_{x_c^t}[\|M_\theta(x_c^t + \lambda^t\tau) - M_\theta(x_c^t)\|]]$. Similar to trigger inversion loss, we can apply $Loss_{rb}$ only at the timestep $T$ and simplify it as: $Loss_{rb} = \mathbb{E}_{\epsilon \sim \mathcal{N}(0,1)}[\|M_\theta(\epsilon + \tau) - M_\theta(\epsilon)\|]$. However, this loss alone is not sufficient, because $M_\theta$ may learn to shift the benign distribution towards the backdoor one instead of the opposite. Therefore, we use the clean distribution of $M_\theta$ on clean inputs as a reference. To avoid interference, we clone $M_\theta$ and freeze the clone's weights. The frozen model is denoted as $M_f$ and $Loss_{rb}$ is changed to: $Loss_{rb} = \mathbb{E}_{\epsilon \sim \mathcal{N}(0,1)}[\|M_\theta(\epsilon + \tau) - M_f(\epsilon)\|]$. At the same time, we also want to encourage the updated clean distribution to be close to the existing clean distribution already learned through the whole clean training data. It can be expressed as: $Loss_{mc} = \mathbb{E}_{\epsilon \sim \mathcal{N}(0,1)}[\|M_\theta(\epsilon) - M_f(\epsilon)\|]$.

With $Loss_{rb} + Loss_{mc}$, we can get $M_{\theta'}$ to invalidate injected backdoor and the ground truth trigger very fast in 20 updates as shown by Figure 6 in the appendix. That is, when we feed the input noise patched with the ground truth trigger, $M_{\theta'}$ won't generate the target image. However, Algorithm 1 can invert another trigger $\tau'$ that can make $M_{\theta'}$ output the target image. A plausible solution is to train with more iterations. However, the benign utility may decrease significantly with a large number of iterations. So we add the original clean training loss $Loss_{dm}$[8] of diffusion models into our backdoor removal procedure. There are two ways to use $Loss_{dm}$. The first way follows existing backdoor removal literature [37, 3, 74, 61], where we can access 10% clean training data. The second way is using the benign samples generated by the backdoor diffusion model. Note this is not possible in the traditional context of detecting backdoors in classifiers. Hence, the complete loss is $Loss_\theta = Loss_{rb} + Loss_{mc} + Loss_{dm}$. Algorithm 5 in the appendix describes our removal method.

## 4  Evaluation

We implement our framework ELIJAH in PyTorch [48]. We evaluate our methods on DMs trained on CIFAR-10 [32] and HQ [25]. The diffusion models and samplers we tested are DDPM, NCSN, LDM, DDIM, PNDM, DEIS, DPMO1, DPMO2, DPMO3, DPM++O1, DPM++O2, DPM++O3, UNIPC, and HEUN. Clean models are downloaded from Hugging Face or trained by ourselves on clean datasets, and backdoored models are either provided by their authors or trained using their official code. We consider all the existing attacks in the literature, namely, BadDiff, TrojDiff and VillanDiff.

**Evaluation Metrics.** We use similar metrics as existing literature [38, 10, 7, 11]: 1) **Detection Accuracy (ACC)** assesses the ability of our backdoor detection method. [9] 2) $\Delta$**FID** measures the relative FID [19] changes between the backdoored model and the fixed one, meaning our effects on benign utility. 3) $\Delta$**ASR** shows the change of ASR, *i.e.*, how well our method can remove the backdoor. ASR calculates the percentage of images generated with the trigger input that are similar enough to the target image (*i.e.*, the MSE w.r.t the target image is smaller than a pre-defined

---

[7]We choose the random forest because it doesn't require input normalization. TV loss value range is much larger than the uniformity score as shown in Figure 4.

[8]This is the vanilla training loss for DMs on clean data. Please refer their original papers for more details.

[9]The training and testing datasets for detection contain *non-overlapping attacks* in the setting where we assume backdoored models are available.

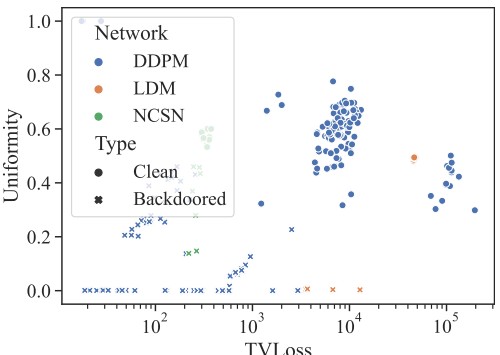

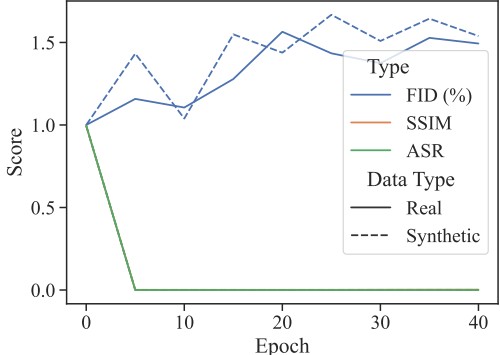

Figure 4: Uniformity scores and TVLoss for 151 clean and 296 backdoored models.

Figure 5: Backdoor removal with only real and synthetic data. ASR and SSIM lines overlap.

Table 1: Overall results of backdoor detection and removal. Model DDPM-C (resp. DDPM-A) means DDPM models trained on CIFAR-10 (resp. CelebA-HQ) dataset. Here ODE-C shows the average results for ODE samplers attacked by VillanDiff. Results of samplers are in Table 5.

| Attack | Model | ACC↑ | ΔASR↓ | ΔSSIM↓ | ΔFID↓ |
|---|---|---|---|---|---|
| Average | | 1.00 | -0.99 | -0.97 | 0.03 |
| BadDiff | DDPM-C | 1.00 | -1.00 | -0.99 | -0.00 |
| BadDiff | DDPM-A | 1.00 | -1.00 | -1.00 | 0.10 |
| TrojDiff | DDPM-C | 0.98 | -1.00 | -0.96 | 0.04 |
| TrojDiff | DDIM-C | 0.98 | -1.00 | -0.96 | 0.03 |
| VillanDiff | NCSN-C | 1.00 | -0.96 | -0.90 | 0.17 |
| VillanDiff | LDM-A | 1.00 | -1.00 | -0.99 | -0.31 |
| VillanDiff | ODE-C | 1.00 | -1.00 | -1.00 | 0.17 |

threshold [11]). A smaller ΔASR means a better backdoor removal. 4) **ΔSSIM** also evaluates the effectiveness of the backdoor removal, similar to ΔASR. It computes the relative change in SSIM before and after the backdoor removal.

**Backdoor Detection Performance.** Our backdoor detection uses the uniformity score and TV loss as the features. Figure 4 shows the distribution of clean and backdoored models in the extracted feature space. Different colors denote different networks. The circles denote clean models while the crosses are backdoored ones. The two extracted features are very informative as we can see clean and backdoored models are quite separable. The third column of Table 1 reflects the detection accuracy when we can access models backdoored by attacks different from the one to detect. Our average detection accuracy is close to 100%, and in more than half of the cases, we have an accuracy of 100%. Our detection performance with only access to clean models is comparable and shown in Table 2.

**Backdoor Removal Performance.** The last three columns in Table 1 show the overall results. ΔASR results show we can remove the injected backdoor completely for all models except for NCSN (almost completely). ΔSSIM reports similar results. With the trigger input, the images generated by the backdoored models have high SSIM with the target images, while after the backdoor removal, they cannot generate the target images. The model utility isn't significantly sacrificed as the average ΔFID is 0.03. For some FIDs with nontrivial increases, the noise space and models are larger (DDPM-A), or the models themselves are more sensitive to training on small datasets (NCSN-C and ODE-C).

**Backdoor Removal with Real/Synthetic Data.** One advantage of backdoor removal in DMs over other models (*e.g.*, classifiers) is we can use the DMs to generate synthetic data instead of requiring ground truth clean training data. This is based on the fact that backdoored models also maintain high clean utility. Figure 5 shows how ELIJAH performs with 10% real data or the same amount of synthetic data. The overlapped SSIM and ASR lines mean the same effectiveness of backdoor removal. The FID changes in a similar trend. This means we can have a real-data-free backdoor removal approach. Since our backdoor detection is also sample-free, our whole framework can work even without access to real data.

**More results in appendix.** Results show our method is robust to different factors (such as the trigger size and poison rate) and adaptive attack cannot succeed.

## Acknowledgment

We thank the anonymous reviewers for their constructive comments. We are grateful to the Center for AI Safety for providing computational resources. This research was supported, in part by IARPA TrojAI W911NF-19-S0012, NSF 1901242 and 1910300, ONR N000141712045, N000141410468 and N000141712947. Any opinions, findings, and conclusions in this paper are those of the authors only and do not necessarily reflect the views of our sponsors.

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

# Appendix

---

**Algorithm 1** Trigger inversion. $T$ is the leftmost step.

---

1: **function** INVERTTRIGGER(model $M$, epoch $e$, lr $\eta$)
2:     Init $\tau$ from $\mathcal{U}[0,1]$
3:     **for** $i \in \{1, \ldots, e\}$ **do**
4:         Sample $\epsilon$ from $\mathcal{N}(0, I)$
5:         $\tau = \tau - \eta \nabla_\tau Loss_\tau(\epsilon)$                                  ▷ Equation (4)
6:     **end for**
7:     **return** $\tau$
8: **end function**

---

---

**Algorithm 2** Extract features for backdoor detection.

---

1: **function** EXTRACTFEATURE(diffusion model $Dm$, img_num $n$, epoch $e$, lr $\eta$)
2:     $\tau = $ INVERTTRIGGER$(Dm.M, e, \eta)$
3:     Sample $x_{b,[1,n]}^T$ from $\mathcal{N}(\tau, I)$
4:     Generate $n$ images $x_{[1,n]} = Dm(x_{b,[1,n]}^T)$
5:     Compute uniformity $s = S(x_{[1,n]})$
6:     Compute TV Loss $l = L_{TV}(x_{[1,n]})$
7:     **return** $(s, l)$
8: **end function**

---

## A   Algorithms

Note that for simplicity, we assume the tensors (*e.g.*, $\epsilon$, $x^T$, etc.) can be a batch of samples or inputs. We only add the indices (*e.g.*, $x_{[1,n]}^T$) when we emphasize the number of samples (*e.g.*, $n$). We call the whole chain a diffusion model denoted by $Dm$ and the learned network model $M$. $Dm(x^T)$ will generate the final image $x^0$ iteratively calling $M$ for $T$ steps.

### A.1   Trigger Inversion

Algorithm 1 shows the pseudocode of our trigger inversion. For a given diffusion model to test, we feed its learned network $M$ to Algorithm 1, and set the epochs and learning rate. Line 2 initializes $\tau$ using a uniform distribution. Line 3-6 iteratively update $\tau$ using the gradient descent on the trigger inversion loss defined in Equation (4). Line 7 returns the inverted $\tau$.

### A.2   Backdoor Detection

Algorithm 2 describes how we extract the uniformity score and TV loss for a diffusion model $Dm$. Line 2 calls Algorithm 1 to invert a trigger $\tau$ for the learned network $Dm.M$. Line 3 samples a set of $n$ trigger inputs $x_{b,[1,n]}^T$ from the $\tau$-shifted distribution $\mathcal{N}(\tau, I)$. Line 4 generates a batch of images $x_{[1,n]}$ by feeding $x_{b,[1,n]}^T$ to $Dm$. Lines 5 and 6 compute uniformity score and TV loss. Line 7 returns the extracted features.

Algorithm 3 shows how we conduct backdoor detection when we have both clean models and backdoored models (attacked by a different method from the one to detect). BUILDDETECTOR takes in a set of clean and backdoored models $\{Dm_i\}$ and the corresponding labels $\{l_i \in \{$ 'c', 'b' $\}\}$ where 'c' stands for clean and 'b' for backdoored. Lines 2-6 extract features for all models and build the training dataset $\mathcal{D}_{train}$ for the random forest. Lines 7-8 train and return a learned random forest classifier $cls$. Given a model to test, DETECTBACKDOOR extracts its features (Line 11) and uses $cls$ to predict its label (Line 12).

When we only have access to the clean models, we use threshold-based detection in Algorithm 4. COMPUTETHRESHOLD computes the thresholds for the uniformity score and TV loss. More specifically, Line 2 initializes the two empty feature lists. Lines 3-6 compute the uniformity score

---

**Algorithm 3** Backdoor detection via random forest.

---

1: **function** BUILDDETECTOR(diffusion models $\{Dm_i\}$, labels $\{l_i \in \{$ 'c', 'b' $\}\}$, img_num $n$,
    epoch $e$, lr $\eta$)
2:    $\mathcal{D}_{train} = \{\}$
3:    **for** $Dm_i, l_i \in \{Dm_i\}, \{l_i\}$ **do**
4:        $f_i = $ EXTRACTFEATURE$(Dm_i, n, e, lr)$
5:        $\mathcal{D}_{train} = \mathcal{D}_{train} \cup \{(f_i, l_i)\}$
6:    **end for**
7:    Train a random forest classifier $cls$ on $\mathcal{D}_{train}$
8:    **return** $cls$
9: **end function**
10: **function** DETECTBACKDOOR(classifier $cls$, diffusion model $Dm$, img_num $n$, epoch $e$, lr $\eta$)
11:    $f = $ EXTRACTFEATURE$(Dm, n, e, lr)$
12:    **return** $cls(f)$
13: **end function**

---

---

**Algorithm 4** Backdoor detection via threshold.

---

1: **function** COMPUTETHRESHOLD(clean diffusion models $\{Dm_i\}$, img_num $n$, epoch $e$, lr $\eta$,
    FPR $\gamma$)
2:    $\mathcal{U}_{train} = [\,], \mathcal{T}_{train} = [\,]$
3:    **for** $Dm_i \in \{Dm_i\}$ **do**
4:        $s_i, l_i = $ EXTRACTFEATURE$(Dm_i, n, e, lr)$
5:        $\mathcal{U}_{train}.\text{append}(s_i)$
6:        $\mathcal{T}_{train}.\text{append}(l_i)$
7:    **end for**
8:    Sort $\mathcal{U}_{train}$ and $\mathcal{T}_{train}$ in ascending order
9:    $\psi_U = \mathcal{U}_{train}[\lfloor |\mathcal{U}_{train}| * \gamma \rfloor]$
10:    $\psi_T = \mathcal{T}_{train}[\lfloor |\mathcal{T}_{train}| * \gamma \rfloor]$
11:    **return** $\psi_U, \psi_T$
12: **end function**
13: **function** DETECTBACKDOORU(diffusion model $Dm$, threshold $\psi$, img_num $n$, epoch $e$, lr $\eta$)
14:    $s = $ EXTRACTFEATURE$(Dm, n, e, lr)[0]$
15:    **return** $s < \psi$
16: **end function**
17: **function** DETECTBACKDOORT(diffusion model $Dm$, threshold $\psi$, img_num $n$, epoch $e$, lr $\eta$)
18:    $l = $ EXTRACTFEATURE$(Dm, n, e, lr)[1]$
19:    **return** $l < \psi$
20: **end function**

---

and TV loss for each clean DM and add them to the lists. Line 8 sorts the two lists in ascending order. Line 9-10 computes the thresholds for uniformity score and TV loss according to the defined False Positive Rate (FPR). Line 11 returns the thresholds. The threshold-based detection algorithms DETECTBACKDOORU and DETECTBACKDOORT are straightforward. Given a model to check, they compute the feature uniformity score (Line 14) or TV loss (Line 18). If the feature is smaller than the threshold, the model is considered backdoored (Line 15 or 16).

## A.3   Backdoor Removal

Algorithm 5 shows the procedure of removing the backdoor. Given a backdoored model $M_\theta$, the inverted trigger $\tau$ and a set of clean (real or synthetic) data $\mathcal{D}$, Line 2 first gets a frozen copy of the backdoored model. Lines 3-9 apply the backdoor removal loop for $e$ epochs. Line 4 gets the clean samples from $D$ as the training samples. Line 5 samples the initial clean noise $x^T$. Lines 6 and 7 compute the backdoored model's outputs of clean inputs and backdoored ones. Line 8 computes the frozen model's outputs of clean inputs as the reference. Line 9 updates $\theta$ using gradient descent on our backdoor removal loss. Line 11 returns the fixed model.

**Algorithm 5** Backdoor removal. $T$ is the leftmost step.

1: **function** REMOVEBACKDOOR(model $M_\theta$, epoch $e$, lr $\eta$, trigger $\tau$, clean data $\mathcal{D}$)
2:     $M_f = \text{FREEZE}(M_\theta)$
3:     **for** $i \in \{1, \ldots, e\}$ **do**
4:         Sample $x^0$ from $\mathcal{D}$
5:         Sample $x^T$ from $\mathcal{N}(0, I)$
6:         $\epsilon_c = M_\theta(x^T, T)$
7:         $\epsilon_b = M_\theta(x^T + \tau, T)$
8:         $\epsilon_f = M_f(x^T, T)$
9:         $\theta = \theta - \eta \nabla_\theta Loss_\theta(\epsilon_c, \epsilon_b, \epsilon_f, x^0)$
10:     **end for**
11:     **return** $M_\theta$
12: **end function**

# B    More Experimental Results

## B.1    Configuration

**Runs of Algorithms.** For each model in the 151 clean and 296 backdoored models, we run Algorithm 1 and Algorithm 5 once, except for Algorithm 3 and Algorithm 4 because clean models and some backdoored models are involved in multiple detection experiments. Given the number of models we evaluated, we believe the results should be reliable.

**Parameters and Settings.** For trigger inversion, we use Adam optimizer [28] and 0.1 as the learning rate. We use 100 epochs for DDPM/NCSN, 10 epochs for LDM and ODE models. We set the batch size to 100 for DMs with $3 \times 32 \times 32$ space, 50 for $3 \times 128 \times 128$ space, and 20 for $3 \times 256 \times 256$ space because of GPU memory limitation. Ideally, a larger batch size will give us a better approximation of the expectation in Equation (4). We set $\lambda = 0.5$ because we tested on a subset of models for $\lambda \in [0, 1]$ with steps 0.1 and found $\lambda = 0.5$ gave the best detection results.

For feature extraction, we only use 16 images generated by input with the inverted trigger since we find it's sufficient.

For the random-forest-based backdoor detection, we randomly split the clean model into 80% training and 20% testing. We add all the backdoored models by one attack to the test dataset. We add all the backdoored models attacked by a different method from the one to test into the training data.

For the threshold-based backdoor detection, we split the clean model into 80% training and 20% testing. We add all the backdoored models by one attack to the test dataset. We derive the thresholds based on the clean training dataset.

To compute $\Delta$ASR, $\Delta$SSIM, and $\Delta$FID, we use 2048 generated images. Generating a lot of images for hundreds of models is very consuming. For example, it take more than 2 hours to generate 2048 $32 \times 32$ samples using NCSN trained on the Cifar10 dataset with batch size 2048 on a NVIDIA Quadro RTX A6000 GPU. Since we are comparing the changes, the trends can be implied by using the same reasonable amount of samples to compare the metric on backdoored model and the corresponding fixed one.

## B.2    Threshold-based Detection

Table 2 compares our detection performance between the random-forest-based method and the threshold-based one with the false positive rate set to 5%. The third column shows the detection accuracy with the random forest. The fourth/fifth column shows the results with a uniformity/TV loss threshold. They perform comparably well while the random forest have a overall higher accuracy.

## B.3    Effect of the Trigger Size

We use TrojDiff to backdoor DMs with various trigger sizes and test ELIJAH on them. Results are shown in Table 3. ELIJAH can detect and eliminate all the backdoors with slight decreases in model utility.

Table 2: Detection accuracy with different settings. ACC means the detection rate with the trained random forest. U@05 means using the threshold extracted on the cleaning training set with a 5% false positive rate. Model DDPM-C (resp. DDPM-A) means DDPM models trained on CIFAR-10 (resp. CelebA-HQ) dataset.

| Attack | Model | ACC(%)↑ | U@5(%)↑ | T@5(%)↑ |
|---|---|---|---|---|
| BadDiff | DDPM-C | 100 | 92.04 | 98.23 |
| BadDiff | DDPM-A | 100 | 100 | 100 |
| TrojDiff | DDPM-C | 98.36 | 100 | 96.36 |
| TrojDiff | DDIM-C | 98.36 | 100 | 96.36 |
| VillanDiff | NCSN-C | 100 | 100 | 100 |
| VillanDiff | LDM-A | 100 | 100 | 100 |
| VillanDiff | ODE-C | 100 | 100 | 98.50 |

Table 3: Performance against different trigger sizes. Numbers show the relative scores compared with backdoored models. The trigger is a white square and the target image is mickey.

| Trigger size | Detected | ΔASR↓ | ΔSSIM↓ | ΔFID↓ |
|---|---|---|---|---|
| 3×3 | ✔ | -1.00 | -0.97 | 0.02 |
| 4×4 | ✔ | -1.00 | -0.97 | 0.04 |
| 5×5 | ✔ | -1.00 | -0.92 | 0.08 |
| 6×6 | ✔ | -1.00 | -0.94 | 0.11 |
| 7×7 | ✔ | -1.00 | -0.94 | 0.05 |
| 8×8 | ✔ | -1.00 | -0.93 | 0.02 |
| 9×9 | ✔ | -1.00 | -0.92 | 0.05 |

## B.4 Effect of the Poison Rate

We evaluate ELIJAH on DMs backdoored by BadDiff with different poison rates. Table 4 demonstrates ELIJAH can completely detect and eliminate all the backdoors and even improve the model utility in many cases.

## B.5 Effect of Clean Data

Figure 6 shows our backdoor removal without clean data can quickly (in 20 updates) invalidate the ground truth trigger so it cannot generate the target image. However, our trigger inversion algorithm can find another effective trigger. With the inverted trigger, the "fixed" model can still generate the target image.

## B.6 Effect of Backdoor Removal Loss

Figure 7, Figure 8a and Figure 8b show fine-tuning the backdoored model only with 10% clean data cannot remove the backdoor. The green dashed line displays the ASR which is always close to 1 for the fine-tuning method, while ours (denoted by the solid green line) quickly reduces ASR to 0 within 5 epochs.

## B.7 Backdoor Removal with Real/Synthetic Data

Figure 9a and Figure 9b show more comparison between backdoor removal with real data and synthetic data. The overlapped SSIM and ASR lines mean the same effectiveness of backdoor removal. The FID changes in a similar trend. This means we can have a real-data-free backdoor removal approach. Since our backdoor detection is also sample-free, our whole framework can work even without access to real data.

Table 4: Performance against different poison rates. Numbers show the relative scores compared with backdoored models.

| Poison rates | Detected | $\Delta$ASR$\downarrow$ | $\Delta$SSIM$\downarrow$ | $\Delta$FID$\downarrow$ |
|---|---|---|---|---|
| 0.05 | ✔ | -1.00 | -1.00 | -0.07 |
| 0.10 | ✔ | -1.00 | -1.00 | 0.18 |
| 0.20 | ✔ | -1.00 | -1.00 | -0.03 |
| 0.30 | ✔ | -1.00 | -1.00 | 0.15 |
| 0.50 | ✔ | -1.00 | -1.00 | 0.20 |
| 0.70 | ✔ | -1.00 | -1.00 | -0.11 |
| 0.90 | ✔ | -1.00 | -1.00 | -0.07 |

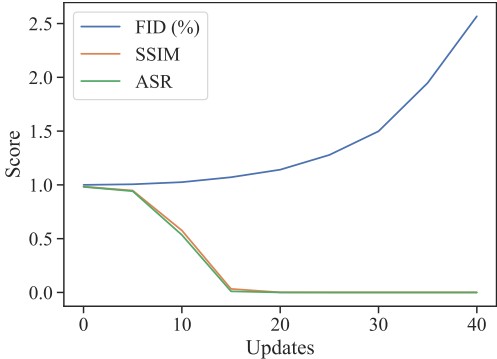 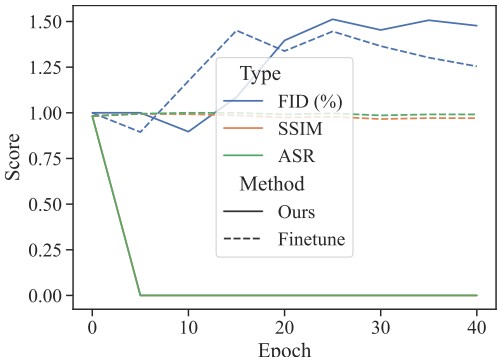

Figure 6: Only use backdoor removal on a model backdoored by BadDiff with stop sign trigger and hat target.

Figure 7: Fine-tuning with only real data cannot remove the backdoor. The model is attacked by BadDiff with stop sign trigger and hat target.

## B.8 Detailed ODE Results

Table 5 shows the detailed results for all the ODE samplers. Our method can successfully detect the backdoored models and completely eliminate the backdoors while only slightly increasing FID.

## B.9 Visualized results

Figure 10 visualizes some ground truth triggers and the corresponding inverted triggers. An interesting observation is that usually, the inverted trigger is not the exact same as the ground truth one. This means the injected trigger is not precise or accurate, that is, a different trigger can also trigger the backdoor effect. It's not an issue for our backdoor detection and removal framework.

## B.10 Results for Inpainting Tasks

BadDiff shows they can also backdoor models used for the inpainting tasks. Given a corrupted image (*e.g.*, masked with a box) without the trigger, the model can recover it into a clean image (*e.g.*, complete the masked area). However, when the corrupted image contains the trigger, the target image will be generated. Our method can successfully detect the backdoored model and completely eliminate the backdoors while maintaining almost the same inpainting capability.

## B.11 Details of Adaptive Attacks

We tried two different ways of adaptive attacks. In both cases, attackers completely know our framework. In the first case, attackers directly utilize our loss the suppress the distribution shift at the timestep $T$. Because the backdoor injection relies on the distribution shift, suppressing it will make the attack fail. In the second case, attackers choose to only inject the distribution shift starting from the timestep $T - 1$ while training the timestep $T$ with only clean training loss. The intuition is our simple trigger inversion loss only uses the timestep $T$. However, this adaptive attack also failed

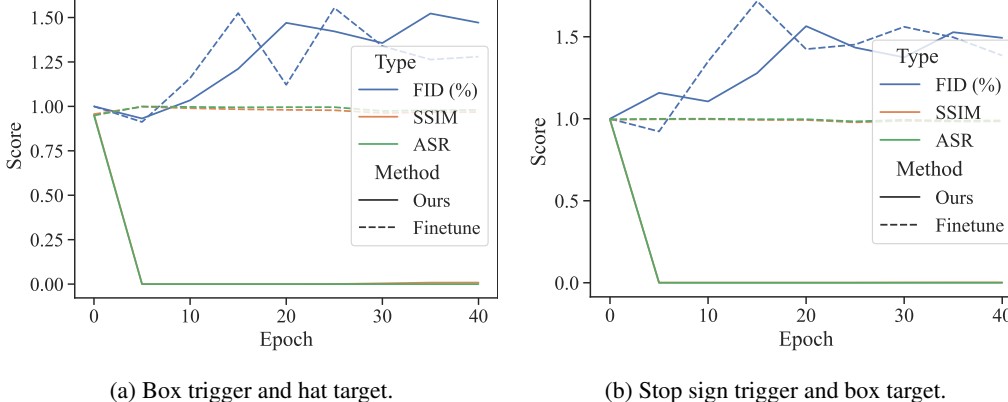

(a) Box trigger and hat target.       (b) Stop sign trigger and box target.

Figure 8: Comparison between ours and fine-tuning with only real data on models backdoored by BadDiff.

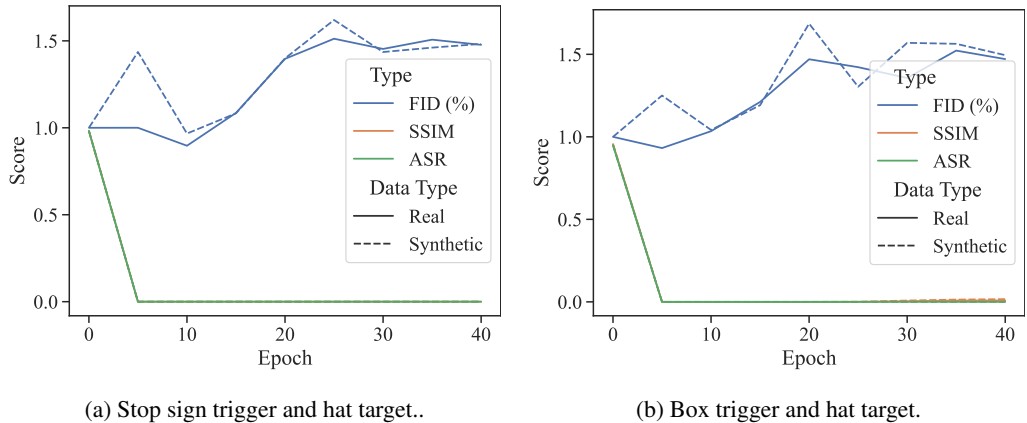

(a) Stop sign trigger and hat target..     (b) Box trigger and hat target.

Figure 9: Backdoor removal with real data or synthetic data on a model backdoored by BadDiff. ASR and SSIM lines overlap.

because even if the $T-1$ step learns the distribution shift, it could not be satisfied by the $T$ step. That is, the attack also failed.

## B.12 Parameter Instantiation for Diffusion Models and Attacks

**DDPM.** This is straightforward, as DDPM is directly defined using a Markov chain. $\kappa^t = \sqrt{\alpha^t}$, $v^t = \beta^t$, $\tilde{\kappa}^t = \frac{1}{\sqrt{\alpha^t}}$, $\hat{\kappa}^t = \frac{1-\alpha^t}{\sqrt{\alpha^t}\sqrt{1-\bar{\alpha}^t}}$, $\tilde{v}^t = \frac{1-\bar{\alpha}^{t-1}}{1-\bar{\alpha}^t}\beta^t$, where $\beta^t$ is the predefined scale of noise added at step $t$, $\alpha^t = 1 - \beta^t$ and $\bar{\alpha}^t = \prod_{i=1}^{t} \alpha^i$.

**NCSN.** $\kappa^t = 1$, $v^t = (\sigma^t)^2 - \sum_{i=1}^{t-1}(v^i)^2$, $\tilde{\kappa}^t = \frac{(\sigma^{t-1})^2}{(\sigma^{t-1})^2+(v^t)^2}$, $\hat{\kappa}^t = 1 - \tilde{\kappa}^t$, $\tilde{v}^t = (1-\tilde{\kappa}^t)(\sigma^t)^2$, where $\sigma^t$ denotes scale of the pre-defined noise.

**LDM.** As LDM is considered DDPM in the latent space, the instantiation is almost the same.

**BadDiff** BadDiff only attacks DDPM, with $\rho^t = 1 - \sqrt{\alpha^t}$, $\tilde{\rho}^t = \frac{(1-\sqrt{\alpha^t})\sqrt{1-\bar{\alpha}^t}}{\alpha^t-1}$.

**TrojDiff** $\rho^t = k^t$, $v_b^t = \beta^t\gamma^2$, $\tilde{\kappa}_b^t = \frac{\sqrt{\alpha^t}(1-\bar{\alpha}^{t-1})}{1-\bar{\alpha}^t} + \frac{1}{\sqrt{\bar{\alpha}^t}}$, $\hat{\kappa}_b^t = \frac{-\sqrt{1-\bar{\alpha}^t}(\gamma}{\sqrt{\alpha^t}}$, $\tilde{\rho}^t = \frac{\sqrt{1-\bar{\alpha}^t}}{\sqrt{\alpha^t}} + \frac{\sqrt{1-\bar{\alpha}^{t-1}}\beta^t - \sqrt{\alpha^t}(1-\bar{\alpha}^{t-1}k^t)}{1-\bar{\alpha}^t}$, where $k^t = \sqrt{1-\bar{\alpha}^t} - \sum_{i=2}^{t}\prod_{j=i}^{t}\sqrt{\alpha^j}k^{i-1}$. Note the subscript shows a different chain for the backdoor from the clean chain.

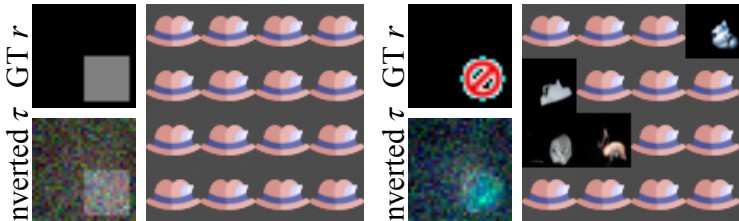

Figure 10: Ground truth triggers $r$ and the corresponding inverted triggers $\tau$, as well as 16 generated images using inputs with the inverted triggers.

Table 5: ODE results of VillanDiff backdoor detection and removal. Model DDIM-C means DM with DDIM sampler trained on the CIFAR-10 dataset.

| Model | ACC↑ | ΔASR↓ | ΔSSIM↓ | ΔFID↓ |
|---|---|---|---|---|
| DDIM-C | 1.00 | -1.00 | -1.00 | 0.14 |
| PNDM-C | 1.00 | -1.00 | -1.00 | 0.25 |
| DEIS-C | 1.00 | -1.00 | -1.00 | 0.15 |
| DPMO1-C | 1.00 | -1.00 | -1.00 | 0.15 |
| DPMO2-C | 1.00 | -1.00 | -1.00 | 0.15 |
| DPMO3-C | 1.00 | -1.00 | -1.00 | 0.15 |
| DPM++O1-C | 1.00 | -1.00 | -1.00 | 0.15 |
| DPM++O2-C | 1.00 | -1.00 | -1.00 | 0.15 |
| DPM++O3-C | 1.00 | -1.00 | -1.00 | 0.15 |
| UNIPC-C | 1.00 | -1.00 | -1.00 | 0.15 |
| HEUN-C | 1.00 | -1.00 | -1.00 | 0.25 |

**VillanDiff** It considers a more general set of DMs and assumes the trigger distribution shift is $\hat{\rho}^t r$. That is, the clean forward chain is $q(x_c^t|x_c^0) = \mathcal{N}(x_c^t; \hat{\alpha}^t x_c^0, \hat{\beta}^t I)$ and the backdoor forward chain is $q(x_b^t|x_b^0) = \mathcal{N}(x_b^t; \hat{\alpha}^t x_c^0 + \hat{\rho}^t r, \hat{\beta}^t I)$. Then $\kappa^t = \frac{\hat{\alpha}^t}{\hat{\alpha}^{t-1}}$, $\upsilon^t = \hat{\beta}^t - \sum_{i=1}^{t-1}((\prod_{j=i+1} t\kappa^j)^2 \upsilon^i)$, $\tilde{\kappa}^t = \frac{\kappa^t \hat{\beta}^{t-1}}{(\kappa^t)^2 \hat{\beta}^{t-1} + \upsilon^t}$, $\hat{\kappa}^t = \frac{\hat{\alpha}^{t-1}\upsilon^t}{(\kappa^t)^2 \hat{\beta}^{t-1} + \upsilon^t}$, $\tilde{\upsilon}^t = \frac{\hat{\kappa}^t}{\hat{\alpha}^t} \hat{\beta}^t$.

### B.13 Related Work

**Diffusion Models and Backdoors.** Diffusion Models have attracted a lot of researchers, to propose different models [20, 56, 57, 26, 58, 53] and different applications [71, 22, 49, 27, 31, 45, 21]. A lot of methods are proposed to deal with the slow sampling process [55, 41, 42, 80, 26, 77] Though they achieve a huge success, they are vulnerable to backdoor attacks [10, 7, 11]. To mitigate this issue, we propose the first backdoor detection and removal framework for diffusion models.

**Backdoor Attacks and Defenses.** When backdooring discriminative models, some poison labels [39, 9, 16] while others use clean label [63, 54]. These attacks can manifest across various modalities [8, 50, 66, 2, 24, 36, 6]. Backdoor defense encompasses backdoor scanning on model and dataset [68, 38, 29] and certified robustness [69, 70, 44, 23]. Backdoor removal aims to detect poisoned data through techniques like data sanitization [43, 60, 14] and to eliminate injected backdoors from contaminated models [37, 3, 74, 76, 33, 34, 67, 73, 30, 62, 65, 61, 35, 72]. These collective efforts highlight the critical importance of defending against backdoor threats in the evolving landscape of machine learning security. However, existing backdoor defense mechanisms and removal techniques have not been tailored to the context of diffusion models.

