# OpenReview forum: "How to remove backdoors in diffusion models?"
_NeurIPS.cc/2023/Workshop/BUGS — NeurIPS 2023 BUGS Poster_

### Official Review · Reviewer_HjyH · 2023-10-27

**Rating:** 7
**Confidence:** 3

**Review:**

The paper proposes the first method to detect and remove backdoors from diffusion models. The main steps of the process are trigger inversion (which assumes a linear dependence in the markov chain, which I am not sure how strong is), followed by backdoor detection which lets them find if a model is backdoored. The paper does not do a good job at explaining this part of the process, and most of it is delegated to the appendix (which is at the reviewer's  discretion to read). The third stage is to clean the model by finetuning on a loss that removes the effect of the trigger detected in the first step. Authors do an extensive experimentation with 300 backdoored models (huge!), several sampling methods, and all backdooring methods for diffusion models, which is really commendable.

Problems with the paper:
1. Table 1 shows the metrics after the cleaning procedure, but miss showing the data before the procedure, making it difficult to buy the claims of having the model cleaned. The metrics ∆ASR is defined as the percentage of images that is close to the target image, I do not understand how can that be negative. The text does not mention if this is the difference between the clean and backdoored model (lines 191 to 198). If it is, that should be clearly mentioned, and the parity direction should be changed to make it more intuitive for the readers.

Minor Comments:
Please move the legend in Figure 4 outside the figure as it is hiding a significant chunk of the points.

---

### Decision · Program_Chairs · 2023-10-28

**Decision:**

Accept (Poster)

**Comment:**

We thank the authors for their thorough work and look forward to their poster!